# The Potential of Private Standards for Valorizing Compliance with Access and Benefit Sharing Obligations of Genetic Resources and Traditional Knowledge

**Hanna Schebesta**

Law Group, Wageningen University & Research, 6706 KN Wageningen, The Netherlands;
Hanna.Schebesta@wur.nl

**Abstract:** International legal instruments such as the Nagoya Protocol on Access to Genetic Resources and the Fair and Equitable Sharing of Benefits Arising from their Utilization to the Convention on Biological Diversity ("Nagoya Protocol") and the International Treaty on Plant Genetic Resources for Food and Agriculture ("ITPGRFA") are meant to create incentives for fairly sharing the benefits of the utilization of genetic resources. These commitments have, however, been assessed by many stakeholders as an obstacle rather than an incentive to commercial activities. If this is indeed the case, ABS obligations may do more harm than good and raises the fundamental question: can ABS obligations be translated from an obstacle into an opportunity? The article discusses consumer-based mechanisms as positive drivers for benefit sharing by using private standards to incentivize ABS obligation compliance. This approach goes further than using private standards as implementation tools, and suggests that they could leverage advantages for industry from the consumer perspective, specifically a consumer-facing label on products. We suggest a research strategy addressing this approach.

**Keywords:** private standards; access and benefit sharing (ABS); Nagoya Protocol

## 1. Introduction

The international governance of genetic resources is addressed by the Convention on Biological Diversity ("CBD") that aims at the conservation of genetic variation, sustainable use and consumption and fair and equitable sharing of benefits arising from genetic resources. Under the CBD, access to genetic resources in a foreign country requires the prior informed consent of the country and the terms and conditions of access and use of this resource must be negotiated and agreed. In order to enhance the fair and equitable sharing of benefits from genetic resources, the Nagoya Protocol (entry into force 2014) set up rules and procedures on access and benefit sharing ("ABS") in compliance with the CBD. Complementary to this, the 2001 International Treaty on Plant Genetic Resources for Food and Agriculture ("ITPGRFA") had already instituted fair and equitable sharing of plant genetic resources. These international frameworks were adopted with a view to reduce perceived global inequalities that accompanied the utilization of genetic resources by users of genetic resources (typically high-income countries) from biodiversity-rich provider countries (often low and middle income countries, "LMICs") [1].

The more recent entry into force of the Nagoya Protocol reignited the debate around ABS of genetic resources and traditional knowledge beyond plant genetic resources. The Nagoya Protocol is being implemented in Europe through Regulation (EU) No. 511/2014 of the European Parliament and of the Council of 16 April 2014 on compliance measures for users from the Nagoya Protocol on Access to Genetic Resources and Fair and Equitable Sharing of Benefits Arising from their Utilization in the Union ("the EU ABS Regulation"). The European Union ("EU") industry and other stakeholders foresee worrying effects on innovation in the sector due to problems in accessing genetic resources, difficult and

complex contract negotiations and protracted administrative tasks. If this is indeed the case, the protocol may do more harm than good and raises the fundamental question: can ABS obligations be translated from an obstacle into an opportunity?

In this article we examine whether compliance with ABS obligations can be translated into an opportunity by using private standards as an instrument for valorizing compliance with such obligations, specifically through a consumer-facing label on products.

In order to create and shape incentives, private regulation can play a role. While the compliance and due diligence effect of private standards has been previously discussed, we argue that a private, consumer-facing standards label helps to integrate the consumer into ABS supply chains and thereby incentivize companies to adhere to ABS obligations.

We first address the legal regulatory framework of ABS obligations (Section 2), and how compliance with ABS obligations, specifically the Nagoya Protocol, in the EU is currently seen as problematic by stakeholders (Section 3). We suggest that a systemic approach should consider not only public regulation, but also the role of private regulation (Section 4). Here, consumer mechanisms, specifically product labels, may provide possibilities for valorization of ABS obligation compliance (discussed in Section 5). We conclude with a recommendation for future research and policy considerations (Section 6).

## 2. The International Legal Framework for the Governance of Genetic Resources

Three main motives guide the international law on genetic resources and traditional knowledge, notably the protection against misappropriation of genetic resources, the conservation of genetic resources and scientific cooperation [2] (p. 2). In this line, the legal objectives of the CBD are the "conservation of biological diversity, the sustainable use of its components and the fair and equitable sharing of the benefits arising out of the utilization of genetic resources, including by appropriate access to genetic resources and by appropriate transfer of relevant technologies, taking into account all rights over those resources and to technologies, and by appropriate funding" (Article 1 CBD). With this understanding, the CBD effected a systemic paradigm shift emphasizing the sovereignty over natural resources [3] (p. 6). Arguably, despite preservation of biodiversity as a prime objective, the CBD embodies a shift towards utilization and economic exploitation of genetic resources, providing a focus on "the monetization of biodiversity" [2] (p. 2). The expectation for ABS is therefore an economic one, which is equally reflected in the Nagoya Protocol that emphasizes the importance of the economic value of biodiversity and sees this as an incentive for conservation [3] (p. 10), (Recital 6 Nagoya Protocol). Additionally, the non-binding Bonn Guidelines on Access to Genetic Resources and Fair and Equitable Sharing of the Benefits Arising out of their Utilization ("Bonn Guidelines") emphasize the role of incentives; in particular advocating well-designed economic and regulatory instruments, the use of valuation methods, as well as the creation and use of markets (Article 51 Bonn Guidelines).

Economic benefits can be generated—and then shared fairly and equitably—if the international ABS regime encourages the use of genetic resources of participating countries in activities such as plant breeding and the development of commercializable products for the bioeconomy such as for medical products. For instance, a study for six African countries concluded that there are promising opportunities for valorization in the sectors of functional foods, cosmetics, pharmaceuticals and biotechnology [4]. However, it is important to bear in mind that the ABS regime will not necessarily increase the use of genetic resources. Some stakeholders view these obligations more as a burden than as a support for using genetic resources; there is the risk that the system disincentivizes the use of genetic resources that fall under ABS regimes.

### 2.1. The CBD

The 1992 CBD was a milestone in biodiversity regulation. The regime strengthens the property rights on genetic resources, and recognizes that countries "own" their biological

resources. The Nagoya Protocol, adopted in 2010 (entry into force in 2014) tops up the CBD by strengthening the access and benefit sharing objective enshrined therein.

The CBD stipulates that each contracting party should take appropriate measures to ensure "sharing in a fair and equitable way the results of research and development and the benefits arising from the commercial and other utilization of genetic resources with the Contracting Party providing such resources." Access requires prior informed consent ("PIC") of the Contracting Party providing access (Article 15(5) CBD). Benefit-sharing obligations rest on the conclusion of mutually agreed terms ("MAT") between user and provider countries (Article 15(7) CBD). The notion of fair and equitable is not closely defined.

Implementing ABS obligations under the CBD soon appeared to be problematic, due to few parties passing ABS legislation, the complexity of the legal issues and the lack of detailed guidance [3] (p. 17). In order to flesh out the CBD provisions on ABS, the non-binding Bonn Guidelines were drawn up in 2002 and adopted by 180 countries. The Guidelines aim to support stakeholders in establishing legislative, administrative or policy measures on ABS and in the negotiation of respective contracts and other arrangements under mutually agreed terms for access and benefit-sharing (Article 1 Bonn Guidelines). In reference to the required MAT under Article 15(7) CBD, the Bonn Guidelines establish basic requirements, an indicative list of typical mutual agreed terms and benefit sharing. Detailing the "how to" of benefit sharing, the Guidelines address the type, timing, distribution and mechanisms for benefit sharing. They also list examples of monetary and non-monetary benefit sharing types (Appendix II of the Bonn Guidelines).

## 2.2. The International Undertaking for Plant Genetic Resources

The CBD had been preceded, under the auspices of the Food and Agriculture Organization ("FAO"), by a different system that was developed for the sustainable and equitable use of plant genetic resources for food and agriculture in 1983 in the form of the International Undertaking for Plant Genetic Resources. The approach was to see genetic resources as a heritage of mankind that should be freely available and exchanged. After the adoption of the CBD, the FAO system was remodeled, resulting in the adoption of the International Treaty on Plant Genetic Resources for Food and Agriculture ("ITPGRFA") in 2001. The ITPGRFA recognizes the sovereign rights of States over their resources and supports the CBD objective for food and agricultural plant genetic resources. ITPGRFA did, however, institute a Multilateral System ("MLS") instituting a pool of genetic resources accessible to everyone for 64 of the most important crops that account for 80 percent of human consumption [5]. Benefits accruing from facilitated access shall be shared fairly and equitably, in particular through exchange of information, access to and transfer of technology, capacity-building and the sharing of monetary and other benefits of commercialization (Article 13.1 ITPGRFA). The Treaty stipulates that access will be made under standard Material Transfer Agreements ("sMTA"), which must include a requirement that the recipient must pay an equitable share of benefits arising from the commercialization of a product to a trust account of the Treaty (Article 13(2)(d)(ii) ITPGRFA). The ITPGRFA Governing Body determines what is an equitable share, i.e., the level, form and manner of the payment, in line with commercial practice (Article 13(2)(d) ITPGRFA).

Despite the envisaged complementarity of the legal instruments, it has been noted that the co-existence of the Nagoya Protocol and the ITPGRFA generates confusion and may bar the sustainable utilization of plant genetic resources [6].

## 2.3. Nagoya Protocol—The Obligations

Shortly after the adoption of the Bonn Guidelines, the international community launched negotiations on an international legally binding instrument that culminated in the adoption of the Nagoya Protocol in 2010.

The Nagoya Protocol, importantly, does not itself establish an access and benefit sharing system; rather it obliges and enables parties to establish an ABS system in their domestic

legal orders. In this, the legally imperfect drafting, such as the lack of "rigorous language and legal consistency checks", of the Nagoya Protocol text has been noted [3] (p. 22). The binding nature of the provisions of the Nagoya Protocol unfolds mostly according to the domestic implementation of the requirements. In terms of substantive ABS requirements, Nagoya Protocol obligations thus arise only in accordance with domestic laws.

The Nagoya Protocol operationalizes the existing CBD objectives on ABS relating to genetic resources, in contrast to the Bonn Guidelines, in legally binding form. Further, it extends benefit-sharing obligations beyond genetic resources to those arising from the use of the traditional knowledge of indigenous and local communities in research and development.

Parties should establish a system under which benefits arising from the utilization, the subsequent applications and commercialization of genetic resources and genetic resources that are held by indigenous and local communities are shared fairly and equitably with the Contracting Party and/or the communities concerned. Sharing should be based on mutually agreed terms (Article 5(1) Nagoya Protocol). The Annex of the Nagoya Protocol provides an example of monetary and non-monetary benefits. Monetary benefits may include, but are not limited to: (a) access fees/fee per sample collected or otherwise acquired; (b) up-front payments; (c) milestone payments; (d) payment of royalties; (e) license fees in case of commercialization; (f) special fees to be paid to trust funds supporting conservation and sustainable use of biodiversity; (g) salaries and preferential terms where mutually agreed; (h) research funding; (i) joint ventures; (j) joint ownership of relevant intellectual property rights, see Annex point 1 Nagoya Protocol. Non-monetary benefits may include, but not be limited to: (a) sharing of research and development results; (b) collaboration, cooperation and contribution in scientific research and development programs, particularly biotechnological research activities, where possible in the Party providing genetic resources; (c) participation in product development; (d) collaboration, cooperation and contribution in education and training; (e) admittance to ex situ facilities of genetic resources and to databases; (f) transfer to the provider of the genetic resources of knowledge and technology under fair and most favorable terms, including on concessional and preferential terms where agreed, in particular, knowledge and technology that make use of genetic resources, including biotechnology, or that are relevant to the conservation and sustainable utilization of biological diversity; (g) strengthening capacities for technology transfer; (h) institutional capacity-building; (i) human and material resources to strengthen the capacities for the administration and enforcement of access regulations; (j) training related to genetic resources with the full participation of countries providing genetic resources, and where possible in such countries; (k) access to scientific information relevant to conservation and sustainable use of biological diversity, including biological inventories and taxonomic studies; (l) contributions to the local economy; (m) research directed towards priority needs, such as health and food security, taking into account domestic uses of genetic resources in the Party providing genetic resources; (n) institutional and professional relationships that can arise from an access and benefit-sharing agreement and subsequent collaborative activities; (o) food and livelihood security benefits; (p) social recognition; (q) joint ownership of relevant intellectual property rights, see Annex point 2 Nagoya Protocol) benefit-sharing types, a list that is vastly inspired by the Bonn Guidelines.

Parties further should establish a system by which access to genetic resources (Article 6 Nagoya Protocol) requires the prior informed consent ("PIC") of the Contracting Party or indigenous and local communities with established rights. The Nagoya Protocol stipulates details of the PIC, but these are left to Contracting Parties to implement as they see appropriate.

Lastly, Parties should establish measures to ensure that access to traditional knowledge associated with genetic resources held by indigenous and local communities is accessed with the PIC or approval and involvement of these indigenous and local communities, and that MAT have been established (Article 7 Nagoya Protocol).

Broadly, the content requirements thus cover access obligations, notably PIC. Parties may require a public permit in order to acquire genetic resources or local traditional knowledge and benefit sharing rules, notably MAT. Parties may require that entities conclude a contract with local partners on how benefits are shared. However, while the Nagoya Protocol establishes more detailed requirements, these are always to be implemented "as appropriate" (Article 5, 6 and 7 Nagoya Protocol), thus giving maximum discretion to countries in implementing their ABS regime.

In contrast to the open norms established for the content of Parties' domestic ABS regime, a legally stringent obligation is created with respect to compliance. While the Nagoya Protocol is weak on rule setting obligations, it institutes a clear duty on Parties to ensure that genetic resources utilized within its jurisdiction have been accessed in accordance with PIC and that MAT have been established, as required by the domestic access and benefit-sharing legislation or regulatory requirements of the other Party (Article 15 Nagoya Protocol).

In addition, the Nagoya Protocol establishes internationally recognized certificates of compliance that serve as evidence that the genetic resource which they cover has been accessed in accordance with PIC and that MAT have been established. Additionally, this rule is applicable only in so far as it is required at the domestic level (Article 17 Nagoya Protocol). A certificate contains information about the parties, as well as the subject matter or genetic resources covered by the certificate; confirmation that MAT were established and PIC was obtained and whether it is for commercial and/or non-commercial use.

Overall, while the content of ABS obligations depends and varies with domestic legal systems, the Nagoya Protocol does create strong compliance requirements.

### 2.4. EU ABS Regulation

The EU ABS Regulation implements the international rules of the Nagoya Protocol in the EU. The Commission Implementing Regulation (EU) 2015/1866 of 13 October 2015 lays down more detailed rules for the implementation of Regulation (EU) No. 511/2014 on the register of collections, monitoring user compliance and best practices ("the Implementing Regulation"). Recently, a specific Guidance Document was published by the European Commission, which intends to clarify the EU rules, notably the concept of "utilization" (Guidance document on the scope of application and core obligations of Regulation (EU) No. 511/2014 of the European Parliament and of the Council on the compliance measures for user from the Nagoya Protocol on Access to Genetic Resources and the Fair and Equitable Sharing of Benefits Arising from their Utilization in the Union 2021/C 13/01).

### 3. The Problem: Nagoya as a Nuisance

For stakeholders, ensuring legal compliance with ABS obligations in their activities is a complex task. Such concerns are particularly valid for the Nagoya Protocol, as it does not institute one harmonized ABS system, and therefore national ABS regulation vastly differ in terms of whether and how Nagoya Protocol requirements are implemented. To date, only 18 countries have notified an ABS procedure [7]. Users of genetic resources, however, have strict compliance obligations. The uncertainty arising from the complexity of ABS requirements, coupled with a strict compliance duty, thus may act as a disincentive for stakeholders to engage in Nagoya Protocol affected supply chains.

In this vein, stakeholders have voiced concern about worrying effects on innovation in the sector due to problems in accessing plant genetic resources, difficult and complex contract negotiations and protracted administrative tasks [1,8].

A recent survey of various sectors (pharmaceuticals, biotechnology, cosmetics, plant breeding, animal breeding, food and agriculture, biocontrol and human health) in the EU showed that interviewees generally found compliance efforts with the Nagoya Protocol disproportionally high and unattainable, on account of the administrative burden, legal uncertainty and lacking information [9]. It was suggested that the EU ABS Regulation should better align the ABS-related obligations with existing sectoral frameworks, and for

instance stakeholders from the food and agriculture sector suggested that all plant genetic resources should be governed under the regime of the ITPGRFA [9].

Concerns about the worrying effect of ABS obligations echo in the literature: "The ABS experience, thus, has not been positive on either end of the equation, for neither provider countries or user countries. Significant benefits have not accrued to the access provider at the level of national implementation from the provision of "access" and the rate of user compliance has been low, partly due to the perception that the rate of return is not commensurate with the efforts required to comply" [10]. Increasingly, the fact that the Nagoya Protocol may have unintended negative consequences is pointed out [for instance, [1,10,11]]. Similarly, a leading United States ("US") attorney in the field has in practice characterized the "quid pro quo" of the ABS process as exceedingly difficult and one that often fails, concluding that it is an "aspirational legal system with systemic flaws" [12] (p. 856). Broadly, the major flaws are found to be in bureaucracy, slowing of research, incompleteness and unclarity and the lack of implementation [13] (pp. 16–19).

As of 2008, only a few successful ABS cases under the CBD had been reported [14]. While new success stories are increasingly emerging (see, for instance a number of case studies with examples of plants for medicinal, cosmetic, biotech and food products in [15]), there are no reliable empirical assessments of the effect of the Nagoya Protocol on ABS practices. Generally, confidentiality clauses in the contractual ABS agreements make it difficult both to provide an accurate empirical assessment of ABS practices, and to assess whether the actual content of agreements is indeed fair and equitable [16]. Qualitative overall evaluations, however, support the view that while investment in ABS is declining, actual benefits (monetary or non-monetary) are only rising insignificantly [16] (p. 8).

The Nagoya Protocol also institutes a "special" mechanism, namely certificates of compliance. At the moment, 23 countries have some form of certificate registered at the Access and Benefit Sharing Clearing House. This number grew exponentially from June 2019 and there are now 2839 registered certificates, for commercial licenses that is. The increase largely came from India, so India is the one player that is really using this certificate of compliance, with 1924 certificates. France is the second major player, with 483 certificates, and the other ones in some way are more negligible.

The role of market-based measures in this appears weak, and "biodiversity-related economic instruments have so far produced 'modest effects at best'" [3] (p. 12). One of the possible explanations for the limited success of ABS is that—apart from the legal obligation to comply with ABS requirements—the actual incentives for companies to take ("good") ABS measures are weak.

For the plant breeding sector, the Nagoya Protocol increases the costs of plant genetic resources registered post-Nagoya. The additional costs arise due to the benefit sharing agreements, and include not only the profit sharing but also the establishment of compliant ABS agreements. Challenges lie, for instance, in identifying the relevant country or countries to arrange agreements with and within the countries the relevant provisions and authorities. Thus far, only 18 Nagoya Protocol signatories notified ABS procedures, and only a few African countries have competent authorities identified for handling the process [6]. Plant breeders have an incentive to only use pre-Nagoya certified plant breeding material to avoid the costs and related uncertainties of using planting material falling under the Nagoya Protocol. From a plant breeders perspective the Nagoya Protocol increases the ex-ante costs of plant breeding, while ex-ante refers to the additional costs before new planting material can be marketed. Legal uncertainty will also still remain considering the vagueness of the access and benefit sharing arrangements.

Based on these legal challenges, users of genetic resources, if faced with a choice of sourcing from a Nagoya-genetic resource or a non-Nagoya genetic resource, may choose to move to non-Nagoya resources. A major question is how to make Nagoya more attractive to the market.

## 4. A Gap in the System: Private Standards as a Way to Overcome Challenges in Ensuring Benefit Sharing in the Supply Chain?

Supply chains that are affected by the Nagoya Protocol lack positive drivers as reported by a number of stakeholders [8,9], and an incentive-based implementation therefore necessitates novel value-generating strategies. Taking into account the objective of the Nagoya Protocol to act as an incentive, the question is how the value that the public may contribute to biodiversity can be translated into an economic value. The parties involved in the implementation of the Nagoya Protocol have realized the possible negative implications. The ABS Clearing House Mechanism has been established explicitly to address legal uncertainty and related costs by supporting Internationally Recognized Certificates of Compliance with the protocol [17]. Nevertheless, obtaining a certificate still bears additional costs and reduces incentives.

In the following, we discuss the role of private regulation in the implementation of ABS obligations (Section 4.1). Standards have an important function for lowering compliance costs. Additionally, we argue that beyond "due diligence", private standards can constitute incentive mechanisms of their own right. We suggest that private standards—if accompanied by a consumer-targeted label, inspired by "fairtrade" labels—can support value creation in relevant supply chains (Section 4.2). Viewed from a "System Perspective", can private standards build a bridge between genetic resource providers and consumers, thus facilitating benefit sharing in the supply chain?

### 4.1. The Nagoya Protocol and the Role of Private Regulation and Private Standards

The importance of private standards in achieving public goals is well established. Private standards are, for instance, the primary mode of governance in global food supply chains [18–20]. Increasingly, a fragmented regulatory space has emerged with governance procedures, such as private standard setting, supplementing more traditional command and control measures by governments [21]. In the market-driven food sector, governments' importance as regulators decreases and is increasingly complemented by private players as standard-setters [22]. However, food safety and quality standards currently dispose of much more efficient private standard systems than social and environmental standards or other private standards with sustainability claims [23] (p. 9).

While the legal ABS obligations are created in public law, and regulated through domestic national ABS regulatory systems, the question remains which role private regulation can have for the implementation of the Nagoya Protocol. A wide understanding of private regulation would broadly cover guidelines, best practices and standards, i.e., "voluntary norms" by private actors.

The Nagoya Protocol text explicitly recognizes the role of voluntary norms, including standards, in giving traction to the obligations contained. Article 20 Nagoya Protocol stipulates that parties shall encourage and take stock of "the development, update and use of voluntary codes of conduct, guidelines and best practices and/or standards in relation to access and benefit-sharing." The EU Regulation, in turn, wants Member States to "encourage the development of sectoral codes of conduct, model contractual clauses, guidelines and best practices" (Article 13(b) of the EU ABS Regulation). "Best practices" (Article 8 of the EU ABS Regulation) are defined as procedures, tools or mechanisms, developed and overseen by associations of users or other interested parties, which help users of genetic resources to comply with the obligations of the EU ABS Regulation. These may be registered in accordance with the Commission Implementing Regulation (EU) 2015/1866 of 13 October 2015 laying down detailed rules for the implementation of Regulation (EU) No. 511/2014 of the European Parliament and of the Council as regards the register of collections, monitoring user compliance and best practices. Thus far, one best practice has been recognized in Commission Decision of 10 May 2019, recognizing the Code of Conduct and Best Practice for Access and Benefit Sharing of the Consortium of European Taxonomic Facilities ("CETAF") as best practice under the EU ABS Regulation (C (2019 3380 final)).

Oliva points to the role that private regulation could play in implementing the Nagoya Protocol [24]. Private standards can be used in order to fill gaps in areas where there is no international law yet, or as a tool to implement requirements in international agreements.

The ISO definition of a standard is "A standard is a document, established by a consensus of subject matter experts and approved by a recognized body that provides guidance on the design, use or performance of materials, products, processes, services, systems or persons." Voluntary–private standards set certain requirements that gain a legally binding status through instruments of private law [25].

Private standards are regarded as complementary to regulatory requirements in two ways: private standards very often (a) serve as a proliferation mechanism that disseminates legal obligations by duplicating existing legal (public law) requirements while endowing these requirements with a more stringent private control and certification system. From this angle there is a particular emphasis on the performance of private standards in terms of ensuring compliance with public law obligations. Certification schemes are recognized as a risk assessment and risk mitigation tools against legal violations [24]. Standards may therefore imply that a compliance liability-shifting effect, where actors are required to comply with a standard, shifts the liability upstream [13,26]. They can have a trade facilitative effect by bridging the gap of different legal systems [13,27]. Private standards also (b) fill gaps left by public law. They may serve to operationalize a given set of open public norms by stipulating more detailed implementing norms, set additional or new "top-up" requirements to the public law minimum, either by extending applicable public law duties to an area that is legally not obliged to cover, or by setting a content requirement that goes beyond public law obligations altogether. In addition, (c) standards have important independent market effects, i.e., business-to-business, they serve as a sign of the level of quality [26] and have important consumer signalling effects, providing e.g., market differentiation [13,28].

### 4.2. Data on Benefit Sharing Clauses in Private Standards

A study conducted in 2011 [29] surveyed the prevalence of biodiversity on the basis of a sample of 36 standards in 2010, i.e., before entry into force of the Nagoya Protocol. Around one third of the private standards reference the Convention on Biodiversity, thus acting as a proliferation mechanism. Only a handful of standards make any further stipulation about the CBD. The exception is BioTrade by the Union for Ethical Biotrade ("UEBT"), which clearly targets CBD compliance in the standard. References to ABS requirements explicitly were even more scarce in the sample: Three standards (the Roundtable on Sustainable Palm Oil ("RSPO"), the Sustainable Agriculture Network ("SAN") and the Environmental and Social policy standard of the European Bank for Reconstruction and Development ("EBRD")) made reference to compensation or the fair and equitable sharing of benefits arising from the utilization of genetic resources. Only two standards (FairWild Foundation and Biotrade) stipulate more detailed ABS rules, such as management plans, compensation for damage, agreements and details on adequate compensation.

A more recent study of 31 sustainability standards [13] found that around half of the surveyed standards reference the CBD and/or the Nagoya Protocol. Of these, only the Union for Ethical Biotrade ("UEBT"), LIFE (fair for life and for life standards) and the Kenya Flower Council directly correlate to the Nagoya Protocol's ABS obligations. From the study, it appears that even within private standards that are specifically dedicated to biodiversity, ABS is not necessarily included in standards' requirements. In the biodiversity specific standard the Climate, Community and Biodiversity standard, biodiversity is understood strictly, and there are references to the CBD, but not to genetic resources. Rainforest Alliance 2017, for instance, does not cover specific ABS obligations. This demonstrates that even within biodiversity private standards, benefit sharing—as a social dimension—is not always included in standards dedicated predominantly to the environmental dimension.

UEBT is specifically dedicated to ethical biotrade and one of the key principles of the standard are the ABS obligations. It lists ABS under principles and has dedicated

provisions on compliance with regulations, including international and national rules on endangered species, ABS and the rights of indigenous peoples and local communities. Above all, it institutes a due diligence system that identifies critical control points of ABS, assesses whether laws are applicable and whether compliance is ensured. When countries have not adopted ABS laws or regulations, voluntary ABS measures should be taken when R&D on the natural ingredient has been undertaken [30].

Generally, most standards contain generic clauses about compliance with national or local laws, thus including national ABS laws based on the Nagoya Protocol. However, with few exceptions, most pertinently UEBT, specific ABS obligations of the Nagoya Protocol are either non-existent or pay mere lip-service to the international agreements.

Currently, therefore, private standards' capacity in supporting an effective implementation of the Nagoya Protocol implementation are largely untapped.

## 5. Can Consumer-Directed Biodiversity Certification Act as a Positive Driver for Benefit Sharing?

In order to interest businesses in partaking in private standards dedicated to ABS, new mechanisms must be created that incentivize companies to engage in private standards. Arguably, due diligence systems on the basis of standards, guidelines and codes of conduct reduce compliance costs for companies [24] (p. 298). From a business point of view, the due diligence approach in private standards mainly creates or amplifies obligations; the reduction in compliance costs may act as one incentive to engage in such private instruments. However, we argue that it is desirable to enhance the correlating incentives of private standards.

One way to achieve value creation in supply chains is through the inclusion of consumer-facing labels. A consumer's product choice nowadays can be a choice for "the good", and a product purchase becomes a political act. Captured under the term "political consumerism" [31], eco or sustainability labels can help to capitalize this trend, as is already the case in the food sector [32]. We therefore propose to extend the idea of the political and ethical consumer beyond food into the ABS arena.

In the food sector, it is already widely recognized that labeling practices favorably inform consumer choice, and businesses see it as an opportunity for a variety of reasons, including market differentiation and price premiums. In the biotrade private standards landscape, however, business-to-consumer standards addressing ABS issues are absent. The main ABS dedicated standard, Ethical BioTrade Standard (UEBT), is largely a business-to-business standard. The UEBT certified logo cannot be used on any products or packaging that is destined to final consumers (Section 1(7) and 2(14) Communications and Claims Policy for UEBT Certification, 2016). An exception is the use of the UTZ logo in the case of herbal teas, where under certain conditions the UTZ certification seal is possible.

However, standard and certification scheme labeling practices have an important effect on consumers [33], and positively influence consumers' purchasing decisions, due to the perception of the level of institutionalization and credibility of labels [34,35].

Importantly, they have also been shown to increase the willingness to pay of consumers. While the exact willingness to pay varies according to products, consumer targets, and methodologies used, there is broad scientific consensus that there is willingness to pay (stated/observed) for ethical products by consumers. In this, the consumer groups Fair Trade/Local/Price sensitive Consumers have differentiated preferences [36]. However, price increases due to a fair trade commitment is perceived as fair by consumers and does not have a negative impact on purchase behavior [37].

There is scientific evidence for willingness to pay (stated/observed) for ethical products by consumers, but only few studies address biodiversity specifically.

Consumer studies have shown that environmental attributes and labels resulted in greater attention [38,39], and higher willingness to pay [40–43]. This is also true for biodiversity, and also that biodiversity ranks slightly lower than other environmental attributes [44–47].

To what extent are the existing standards and labels appropriate for promoting biodiversity-friendly production and commercialization? Ninety-three percent of surveyed persons would be more interested in buying from a company that pays attention to biodiversity [48]. At the same time, 83% of consumers claimed they had heard of biodiversity but only 39% provided a correct definition [49].

A study conducted by GIZ [50] found that too few mechanisms exist to inform consumers of the impacts of the implementation of best practices and to promote a change in consumer purchasing behavior towards products that are more sustainable or biodiversity-friendly. For successful biodiversity-friendly production and commercialization, the study recommends to use multiple communication methods in addition to labels.

Nunes and Riyanto [51] identified three conditions for a willingness to pay premium for certification and ecolabeling policy practices: consumers are aware of the non-market biodiversity benefits; consumers are able to internalize biodiversity benefits and the production costs are not too sensitive to the certification schemes. Still, the possibility that the product differentiation created by certification can lead to an increase in the demand of both biodiversity oriented the market segment and the remaining one.

On this basis, we argue that private standards that incorporate a consumer-facing label, inspired by "fairtrade" labels, can possibly generate consumer-based incentives for businesses to actively pursue ABS and thereby rejuvenate the market-based instruments for the implementation of the Nagoya Protocol.

## 6. Valorizing Compliance with ABS Obligations: The Potential of Private Standards for Access and Benefit Sharing

The international regime on genetic resources is based on a liberal environmentalist rationale; it strengthens the proprietary utilization of genetic resources and traditional knowledge, the benefits of which should then accrue fairly and equitably to the providers, including respective communities. In this sense, more utilization would mean more benefits to share. If stakeholders decide to avoid genetic resources falling under the Nagoya Protocol in their activities, then the potential to create benefits for provider countries and communities is not realized.

As shown, the Nagoya Protocol is politically a compromise text, leaving the implementation to the domestic systems. Legally, this results in a complex web of different legal regimes that are often not fully functional and are themselves wrought with legal uncertainties.

There is ample evidence from stakeholders that they perceive the Nagoya Protocol as an obstacle, and that the actual incentives for companies to engage in ABS measures are weak. Therefore, users of genetic resources may choose to move to non-regulated resources if faced with a choice.

In order to address this problem and make ABS compliance more attractive to the market, we propose to reinvigorate the role of private standards. Standards can disseminate international legal instruments' requirements and lower compliance costs. Additionally, we argue that consumer-facing private standards with a label can support value creation in relevant supply chains.

This proposal would need to be substantiated in further research that explores the practical feasibility in different supply chains. With a view of testing the proposal and making an initial exploratory assessment of the possibility of private standards to support ABS obligations, we conducted an exploratory workshop with stakeholders (18 December 2019, WUR campus with participants that were experts and stakeholders from academia (consumer behavior, supply chain, legal), institutional actors (European Commission, national focal point), standard bodies, and industry (cosmetics, plants, pharma, food)). On this basis, we suggest a research strategy to develop the possibilities for valorization of ABS obligations through consumer-facing private labels on three aspects.

- The recommendation for an ABS label is based on the assumption that consumers are, indeed, willing to pay a price premium for a "benefit sharing label". This assumption

requires further study, particularly in light of the inflation of product labels: *Are consumers in Europe willing to pay a higher price for products labelled as being certified for complying with ABS obligations? Which are the supply chains or product groups which lend themselves to ABS labels on products?*

- Even if a consumer willingness to pay can be shown, it is necessary to study and identify the conditions and willingness of supply chain actors and relevant policy makers to participate in such a certification scheme: *Which are the supply chains or product groups with the highest potential?* Specific sectors (e.g., crops in food and agriculture, cosmetics) and specific types of genetic resources (e.g., microbial, animal and forest genetic resources) have their own characteristics that need to be further studied.

- Lastly, the design of the standard will have to be carefully considered. It is necessary to study the most effective way of using private standards for ABS compliance in particular to compare multidimensional standards and biodiversity specific standards, their relationship to public standards, in particular organics, and what kind of requirements should be included: *What are the important design characteristics that need to be considered and will this in the end be feasible? Which requirements should be included?*

These research questions can help to assess the possibilities for translating compliance with ABS obligations from an obstacle into an opportunity. The investigation of the prospects for a labeled standards shows some promise in order to create additional incentives for companies, and thereby give greater traction to international ABS obligations.

This research could further feed into the EU Green Deal, specifically the EU Biodiversity Strategy for 2030. Bringing nature back into our lives that argues that biodiversity considerations need to be better integrated into public and business decision-making and commits to develop methods, criteria and standards to describe the essential features of biodiversity, its services, values, and sustainable use. It is telling that the document does not mention the Nagoya Protocol or the ITPGRFA. To date, the Green Deal is focused on measuring environmental footprints (in a strict sense) and could be enlarged to encompass consideration of the social dimension of biodiversity utilization as enshrined in the international ABS regimes.

**Funding:** This research received no external funding. The organization of the Nagoya Protocol Research Workshop was funded by an Excellence Grant of Wageningen University & Research.

**Institutional Review Board Statement:** Not applicable.

**Informed Consent Statement:** Not applicable.

**Data Availability Statement:** Not applicable.

**Acknowledgments:** The author is grateful to all participants of the Nagoya Protocol Research Workshop for their valuable thoughts and contributions.

**Conflicts of Interest:** The author declares no conflict of interest.

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
