# Peer review of "The Potential of Private Standards for Valorizing Compliance with Access and Benefit Sharing Obligations of Genetic Resources and Traditional Knowledge"

_agronomy, doi:10.3390/agronomy11091823_

Round 1

Reviewer 1 Report

This is an interesting and original piece of work. However, while the author's key proposal is theoretically sound, it is not really certain if it would be feasible in practice. Nevertheless, the author seems to admit this in section 6 of the paper. Clearly, there is need for more research into whether this proposal can actually work in practice.

I would have recommended that the paper be accepted in its present form but there are some minor corrections that may be helpful (e.g. the use of 'PIN' instead of 'PIC' on page 3 of the paper). I would suggest that the paper be properly proofread prior to publication.

Author Response

Comments and Suggestions for Authors

This is an interesting and original piece of work. However, while the author's key proposal is theoretically sound, it is not really certain if it would be feasible in practice. Nevertheless, the author seems to admit this in section 6 of the paper. Clearly, there is need for more research into whether this proposal can actually work in practice.

Indeed. Clarified this point in section 6: a) The article does not provide clarity about whether the suggestion is feasible in (business) practice; since it is an article written from the legal discipline, we explored and strengthened the legal feasibility and the possibility of supporting goals of the legal framework. b) further research is necessary. 

I would have recommended that the paper be accepted in its present form but there are some minor corrections that may be helpful (e.g. the use of 'PIN' instead of 'PIC' on page 3 of the paper). I would suggest that the paper be properly proofread prior to publication.

Well spotted. Corrected it, thank you. Also proofread additionally.  

Reviewer 2 Report

The paper is meant to be published on the Special Issue "The Role of Policies in Plant Breeding—Rights and Obligations" and should therefore be focused on topics related to Access and Benefit Sharing to Plant Genetic Resources for Food and Agriculture, which are relevant for plant breeding, variety obtention and protection and seed production and trade. In fact, the paper correctly includes the International Treaty on Plant Genetic Resources for Food and Agriculture (ITPGRFA) between the international instruments governing the Access and Benefit Sharing to Genetic Resources (see section 2 "The Legal Framework for the Governance of Genetic Resources"). The paper does not discuss the overlapping between the Nagoya Protocol and the ITPGRFA, which generates confusion and in many cases bars the sustainable utilisation of plant genetic resources (see for instances Halewood, M. (2015) (https://www.bioversityinternational.org/e-library/publications/detail/mutually-supportive-implementation-of-the-plant-treaty-and-the-nagoya-protocol/). The proposal of development of consumer-mechanisms to incentivate Nagoya Protocol compliance should consider also the connection with ITPGRFA and better centered on plant genetic resources for food and agriculture, which are the object of the special issue of Agronomy. 

In addition, the description of the current product labels, miss to consider the forest genetic resources, which are of great relevance for conservation and sustainable use of biodiversity.

It would be advisable to better explain the methodology used for the facilitation of the exploratory stakeholder workshop held in 2019, or at least cite a reference where the workshop is reported. 

Author Response

The paper is meant to be published on the Special Issue "The Role of Policies in Plant Breeding—Rights and Obligations" and should therefore be focused on topics related to Access and Benefit Sharing to Plant Genetic Resources for Food and Agriculture, which are relevant for plant breeding, variety obtention and protection and seed production and trade. In fact, the paper correctly includes the International Treaty on Plant Genetic Resources for Food and Agriculture (ITPGRFA) between the international instruments governing the Access and Benefit Sharing to Genetic Resources (see section 2 "The Legal Framework for the Governance of Genetic Resources"). The paper does not discuss the overlapping between the Nagoya Protocol and the ITPGRFA, which generates confusion and in many cases bars the sustainable utilisation of plant genetic resources (see for instances Halewood, M. (2015) (https://www.bioversityinternational.org/e-library/publications/detail/mutually-supportive-implementation-of-the-plant-treaty-and-the-nagoya-protocol/). The proposal of development of consumer-mechanisms to incentivate Nagoya Protocol compliance should consider also the connection with ITPGRFA and better centered on plant genetic resources for food and agriculture, which are the object of the special issue of Agronomy. 

The paper discusses compliance with ABS obligations and the potential of private standards to valorise such fair utilisation of genetic resources. The reviewer observes very correctly that the ABS obligations do not uniquely arise from the Nagoya Protocol/CBD, but in specific cases from the ITPGRFA.

  • We have revised the text accordingly, and refer to compliance with ABS obligations rather than compliance with the Nagoya Protocol where necessary.
  • In addition, we followed the advise of clarifying the legal overlap between the ITPGRFA and the NP, both in the section of the legal framework, as in the stakeholder perception.
  • We also included more attention to the affect on different sectors, specifically food and agriculture, and for different supply chains.

In addition, the description of the current product labels, miss to consider the forest genetic resources, which are of great relevance for conservation and sustainable use of biodiversity.

We have noted the need to consider the specificity of sectors strongly in future research about the role of product labels. 

It would be advisable to better explain the methodology used for the facilitation of the exploratory stakeholder workshop held in 2019, or at least cite a reference where the workshop is reported. 

Good point. Included the link to where the workshop is reported. 

Round 2

Reviewer 2 Report

The author has substantially addressed my comments to the previous version. The paper is now better balanced and more focused on plant genetic resources utilisation for plant breeding, variety obtention and protection and sed production and trade.  ABS are correctly considered as arising from Nagoya Protocol and the International Treaty.